# Design and Characterization of Bispecific and Trispecific Antibodies Targeting SARS-CoV-2

**DOI:** 10.3390/vaccines13030255

**Published:** 2025-02-28

**Authors:** Jiayang Wang, Qi Qian, Yushan Jiang, Zuxin Liang, Yun Peng, Wei Zhao, Yang Yang, Chenguang Shen

**Affiliations:** 1BSL-3 Laboratory (Guangdong), Guangdong Provincial Key Laboratory of Tropical Disease Research, School of Public Health, Southern Medical University, Guangzhou 510515, China; steven_990811@126.com (J.W.); yushan.y.jiang@gmail.com (Y.J.); liangzx2014@163.com (Z.L.); zhaowei@smu.edu.cn (W.Z.); 2National Clinical Research Center for Infectious Disease, Shenzhen Third People’s Hospital, Second Hospital Affiliated to Southern University of Science and Technology, Shenzhen 518112, China; 18856226213@163.com (Q.Q.); yun_peng403@163.com (Y.P.); 3Department of Laboratory Medicine, Zhujiang Hospital, Southern Medical University, Guangzhou 510515, China; 4Key Laboratory of Infectious Diseases Research in South China, Southern Medical University, Ministry of Education, Guangzhou 510515, China

**Keywords:** COVID-19, SARS-CoV-2, bispecific antibodies, trispecific antibodies

## Abstract

Background/Objectives: COVID-19, caused by SARS-CoV-2, has emerged as a global pandemic since its outbreak in 2019. As an increasing number of variants have emerged, especially concerning variants such as Omicron BA.1, BA.2, XBB.1, EG.5, which can escape the immune system and cause repeated infections, they have exerted significant pressure on monoclonal antibodies and the treatment approaches for COVID-19. Broad spectrum antiviral medication was urgently needed. In this study, we developed several bispecific antibodies based on the IgG-scFv format and one trispecific antibody containing Fab fragments with different anti-virus mechanisms studied previously. The Fab fragments are from h11B11, S2P6, and S309 respectively. Method: all recombinant antibodies were expressed by HEK 293. The pseudoviruses’ neutralization assay and the virus challenge to BALB/c mice were deployed to assess the efficiency of recombinant antibodies in vitro and in vivo. Results: the bispecific antibodies exhibited a favorable pseudoviruses neutralization activity, with IC_50_ values ranging from 8 to 591 ng/mL. The trispecific antibody performed even better, with IC_50_ values ranging from 5 to 27 ng/mL. Furthermore, the virus challenge to mice confirmed that the bispecific antibodies, including the trispecific antibody, had decent therapeutic efficacy. Conclusions: our study provided several supplements to the therapeutic measures of COVID-19 based on multispecific antibodies, supporting the great potential of the multispecific antibodies strategy in dealing with emerging pathogens.

## 1. Introduction

The outbreak of COVID-19, which was caused by SARS-CoV-2 in 2019, has passed five years now. Although the World Health Organization (WHO) declared on 5 May 2023 that COVID-19 no longer constituted a public health emergency of international concern (PHEIC), SARS-CoV-2 has not disappeared. According to the WHO, SARS-CoV-2 has led to 775,673,955 reported cases and 7,053,524 reported death cases [1,2]. SARS-CoV-2 is categorized as sarbecoviruses, with a positive-sense single-stranded RNA at the core [3]. The genome of SARS-CoV-2 encodes a series of structural proteins, such as the spike (S) protein, nucleocapsid (N), membrane (M) protein, and envelop (E) protein, which are responsible for invading host cells and initiating immune reactions [4]. The S protein can be divided into S1 and S2 subunits, and the S1 subunit possesses the receptor-binding domain (RBD) that recognizes the human angiotensin-converting enzyme 2 (hACE2) as the target protein [5]. The S2 subunit exerts a role in the prefusion structure. Once bound to hACE2, the S protein undergoes double cleavage to transform into the prefusion structure. The first cleavage site is situated at the boundary of S1 and S2, which is cleaved by furin, while the second site is located at the S2′ site of the S2 subunit, cleaved by proteases. In this manner, the S2 subunit anchors the S protein to the virion membrane and mediates membrane fusion [6].

The infection of SARS-CoV-2 may result in severe pneumonia, which could lead to death or deteriorate prognosis, especially for those patients with risk factors such as advanced age, hypertension, cardiovascular disease, obesity, high BMI, diabetes, and immunosuppression [7,8,9,10,11], and it may have a worse effect on child patients [12]. The treatment of COVID-19 can be classified into two parts: one is anti-inflammatory therapy, and the other is antiviral therapy [13]. During the pandemic, a multitude of anti-inflammatory medications were administered to patients in clinical settings, which include nonsteroidal anti-inflammatory drugs (NSAIDs) [14], glucocorticoids [15], and anti-inflammatory antibodies like tocilizumab and sarilumab [16,17]. Anti-virus medications include remdesivir [18], paxlovid (nirmatrelvir/ritonavir) [19], sotrovimab [20], casirivimab/imdevimab [21], amubarvimab/romlusevimab [22], and azvudine [23,24]. Nevertheless, some medications have certain limitations that restrict the optimal administration timing, such as paxlovid. It is recommended that the drug be administered within 5 days of symptom onset to adult patients with high-risk factors [25]. The antibody drugs are confronted with the challenge posed by evolving viral variants, which might reduce their efficacy [26,27]. Consequently, there is an urgent need to develop a broad-spectrum therapeutic agent or antibody for strategic future stockpiling.

The bispecific antibody (bsAb) is a type of recombinant protein encompassing two distinct antigen-binding regions. The structural diversity of bsAbs is dictated by the specific localization of the second binding domain within their molecular architecture. The format that positions the variable regions of heavy and light chains to the Fab fragment of another antibody using a peptide linker is termed the dual-variable domain (DVD-Ig). The structure that links the second binding region as a single-chain fragment variable (scFv) to the terminus of the heavy chain of another antibody is named IgG-scFv. Additionally, there exists a format that integrates two non-homologous heavy and light chain pairs into a single antibody, resulting in an asymmetrical configuration known as a crossMab [28]. Since the bsAb was first described in the 1960s and generated in the 1980s, there have been numerous applications of bsAbs in the treatment of cancers, such as blinatumomab for leukemia [29] and amivantamab for lung cancer [30]. Meanwhile, there are bispecific antibodies (bsAbs) for infectious diseases in the research stage, such as bsAb15 for SARS-CoV-2 [31], a Fc fusion bispecific antibody against H5N1 influenzas virus [32], and a humanized bispecific antibody (Bis-Hu11-1) against influenza A (H1N1) pdm09 virus [33]. Based on previous studies, the bsAbs exhibit a broader anti-virus spectrum, indicating their potential as promising therapeutic agents in future clinical applications.

In this study, several bsAbs based on the IgG-scFv format and a trispecific antibody were developed. We selected three monoclonal antibodies that have been previously investigated to construct the variable fragments of bsAbs. They are h11B111, S2P6, and S309. Previous studies have demonstrated that ursodeoxycholic acid (UDCA) could decrease the sensitivity of corresponding tissues to SARS-CoV-2, even in the face of the Omicron variant, by downregulating hACE2, suggesting that hACE2 could be a site for the development of an anti-virus medication [34,35]. The h11B11 is a humanized antibody targeted at hACE2 without influencing the carboxypeptidase activity of hACE2 in vitro. The mechanism of h11B11 involves binding to hACE2 with a buried surface area, which blocks the recognition of the target protein by the SARS-CoV-2 spike protein, thereby protecting the host cells [36]. S protein is a crucial component of SARS-CoV-2 that plays a role in initiating the infection process to the host cell, particularly the RBD in the S1 subunit. Therefore, many neutralizing antibodies are targeted at RBD. However, immune escape variants of SARS-CoV-2 emerged, which have frequent mutations on the S1 subunit [37] and would render neutralizing antibodies ineffective and cause repeated infection. On the contrary, the S2 subunit possesses more conserved sites and is capable of eliciting antibodies against various variants of SARS-CoV-2, which would possess a broader anti-virus potential [38]. S2P6 is an antibody isolated from COVID-19 patients, targeting the stem helix (SH region) of the S2 subunit with residues ranging from 1146 to 1159. The S2P6 binds to the target sites through shape-complementarity and hydrogen bonding, involving the complementarity-determining regions (CDRs) of the entire heavy chain (H1 to H3) and CDR1 and CDR3 of the light chain [39]. 1. It possesses a broad spectrum of activity against beta-CoVs such as SARS-CoV-2 and SARS-like viruses, including MERS-CoV, HCoV-HKU1, and HCoV-OC43 [40]. Although the mutations of RBD are the primary cause of immune escape variants, there are still certain antibodies targeting RBD with a broad antiviral spectrum. S309 is one of them, which is isolated from a patient recovering from SARS and possesses the ability to bind to the RBD of SARS-CoV-2. S309 would engage epitopes without blocking the hACE2 upon binding to the S protein but would cause S protein trimer cross-linking, resulting in the hindrance or aggregation of virions [41]. Meanwhile, the epitopes recognized by S309 are conserved in both SARS-CoV-1 and SARS-CoV-2, resulting in the broad-spectrum efficiency of S309 [40].

Based on the aforementioned known research, we have developed several bispecific antibodies and a trispecific antibody, and evaluated the binding activity and protective efficiency at both the cellular and individual levels. These experiments will determine whether these recombinant antibodies have the potential to serve as adjunctive therapeutic agents in the treatment of COVID-19.

## 2. Materials and Methods

### 2.1. Antibodies Expression and Purification

All the variable region sequences of native antibodies were analyzed by VBASE2 [42]. The peptide linker in this study were (GGGGS)_3_ linked the variable regions of scFv or (GGGGS)_9_ linked the scFv to the constant region. The expression vector used in the study was pCAGGS with penicillin resistance. The bispecific antibodies, trispecific antibodies and native antibodies were all expressed as soluble proteins in HEK293F cells with the culture conditions of 37 °C, 5% CO_2_, 150 rpm. The plasmids of the heavy chain and light chain of each antibody were transfected into cells using polyethyleneimine (PEI) at a 1:3 mass ratio. Before transfection, the plasmid and PEI were mixed and incubated at 37 °C for 15 min to facilitate complex formation. Cells were collected and separated by centrifugation at 5000 rpm for 20 min to pellet the cells and obtain a clear supernatant. To ensure the complete removal of cellular debris, the supernatant was then filtered through a 0.45 μm pore-size membrane with a syringe. The clarified supernatant was subsequently purified using Protein A/G affinity columns. The wash buffer and elution buffer were added successively, and the elution buffer was collected. The obtained elution buffer was transferred into a dialysis bag in pre-cooled 1× phosphate-buffered saline (PBS) dialysate and stirred gently at 4 °C on a magnetic stirrer. The dialysate was replaced at 2–4 h, 6–8 h and 10–14 h, respectively. All purified antibodies were analyzed by SDS-PAGE to gauge the purity and molecular weight size.

### 2.2. ELISA Assay

The ELISA plates were coated with antigen protein at a concentration of 1 ng/μL per well, with a volume of 100 μL per well, and incubated overnight at 4 °C. Subsequently, the plates were transferred to an automatic plate washer and washed three times using PBS containing 0.05% *v*/*v* Tween-20 (PBST). To block non-specific sites, 200 μL of PBS with 2.5% *w*/*v* nonfat dry milk was added to each well and incubated at 37 °C for 3 h, followed by repeating the washing steps. The purified antibodies were added to the first well at a concentration of 0.1 μg/μL (recombined antibodies) or 0.05 μg/μL (original antibodies) against their specific antigen as the primary antibody, then subjected to 5 times gradient dilution from the first well to the last well and incubated at 37 °C for 30 min, followed by repeating the washing steps. Next, the horseradish peroxidase (HRP)-conjugated goat anti-human IgG was mixed into PBS with 2.0% *w*/*v* nonfat dry milk at a volume/volume ratio of 1:500, and 100 μL of the solution was added to each well, incubated at 37 °C for 30 min, followed by repeating the washing steps. Next, 100 μL of tetramethylbenzidine (TMB) substrate (WANTAI BioPharm, Beijing, China) was added to each well at room temperature in the dark for 10 min. The reaction was terminated with a 2M H_2_SO_4_ solution, and the absorbance was measured at 450 nm.

### 2.3. Pseudoviruses Neutralization Assay

SARS-CoV-2 pseudoviruses were produced by co-transfecting HEK-293T cells (ATCC, Manassas, VA, USA) with human immunodeficiency virus backbones expressing firefly luciferase (pNL4-3R-E-luciferase) and pcDNA3.1 (Invitrogen, Waltham, MA, USA) expression vectors encoding either wild-type or mutated S proteins. Viral supernatant was harvested 48 h later. Pseudoviruses were incubated with serial dilutions of nAbs at 37 °C for 1 h. Hela-hACE2 cells expressing human ACE2 protein were subsequently added in duplicate to the mixture. The percentages of antibody neutralization were determined by measuring luciferase activity in relative light units (Bright-Glo Luciferase Assay Vector System, Promega Bioscience, Madison, WI, USA) 48 h after exposure to the virus-antibody mixture using GraphPad Prism 7 (GraphPad Software Inc., Boston, MA, USA).

### 2.4. Therapeutic Efficacy Test in Mice

Eight-week-old female BALB/c mice were utilized for this study, as previously reported for Omicron infection experiments in BALB/c mice [43]. All animals were randomly allocated to the treatment groups, each group had 5 mice, and all in vivo experiments were ratified by the Ethics Committee of the Laboratory Animal Center of Shenzhen Third People’s Hospital. All operations involving live SARS-CoV-2 were carried out in the Biosafety Level 3 (BSL3) Laboratories. Mice were mildly anesthetized with isoflurane and inoculated intranasally with 5 × 10^5^ TCID_50_ of XBB.1.16 in 75 μL DMEM 12 h before antibody protection experiments. The antibodies and control IgG (an antibody targeted at hemagglutinin) were injected intraperitoneally. Three days after the virus challenge, lung tissues were collected to quantify the viral load by qPCR by determining the N gene fragment from SARS-CoV-2.

### 2.5. Surface Plasmon Resonance

Surface plasmon resonance (SPR) analysis was performed with the PlexArray HT A100 (Plexera, Woodinville, WA, USA). The antigen proteins were loaded on the 3D Dextran chip. A pH2.0 glycine-hydrochloric acid buffer was used as regeneration solution.

### 2.6. Statistic Analysis

The one-way ANOVA, Dunnett’s multiple comparisons test, and Tukey’s multiple comparisons test were used in this study to assess the in vivo and in vitro experiment by GraphPad Prism 8.0.

## 3. Results

### 3.1. The Design of Recombinant Bispecific Antibodies

The bsAb is a type of recombinant protein capable of recognizing two targeted antigens. In previous studies, bsAb has demonstrated a higher efficiency than normal monoclonal antibodies in the treatment of SARS-CoV-2 [31,32,33]. There exist numerous structural designs for bsAb, among which some are based on scFv, while the rest are based on the entire chain of IgG. In this study, we selected the IgG-scFv structure for designing the recombinant bsAb and modified the positions of the two antigen-binding fragments to assess whether the different positions would impact the binding activity and protective efficacy. The IgG-scFv structure attaches a scFv targeting another antigen to the terminus of the IgG heavy chain. The variable regions of bsAb are designed based on three previously identified monoclonal antibodies, namely h11B11, S2P6, and S309 (Figure 1a–g). The h11B11 is targeted at hACE2, which is also the target protein of SARS-CoV-2 [36]. The S2P6 is aimed at the S2 subdomain of the S protein of SARS-CoV-2 [39]. The S309 is targeted towards the receptor binding domain (RBD) of the S protein [41]. The trispecific antibody is engineered to incorporate three variable regions of monoclonal antibodies above. All recombinant antibodies and original monoclonal antibodies are expressed by HEK293F and soluble proteins could be obtained.

### 3.2. The Recombinant Antibodies Effectively Bind to Specific Antigens

After obtaining and purifying all bispecific antibodies and original antibodies, dialysis was utilized to replace the solvent with PBS. An ELISA test was conducted to evaluate the binding activity of each antibody to its specific antigen. Here, we present the ELISA results by different antigens. Regarding hACE2, the half-maximal effective concentration (EC_50_) of the original monoclonal antibody h11B11 is 29.98 ng/mL, the EC_50_ of the recombinant bsAb F-h11B11 + S2P6, F-h11B11 + S309, F-S2P6 + h11B11, and F-S309 + h111B11 are 98.65, 84.66, 196.5, and 102.5 ng/mL, respectively (Figure 2a). Regarding the S2 protein, these antibodies exhibit a similar trend to those targeting hACE2. The original monoclonal antibody S2P6 demonstrates superior activity against the antigen, with an EC_50_ value of 11.96 ng/mL. Meanwhile, the EC_50_ values of the recombinant bsAbs F-h11B11 + S2P6, F-S2P6 + S309, F-S2P6 + h11B11, and F-S309 + S2P6 are 62.11, 68.39, 51.69, and 97.16 ng/mL, respectively (Figure 2b). For those antibodies targeting XBB RBD, the EC_50_ of the native monoclonal antibody S309 is 118.4 ng/mL, the EC_50_ values of the bsAbs F-S2P6 + S309, F-h11B11 + S309, F-S2P6 + S309, F-H11B11 + S309, F-S309 + h11B11, and F-S309 + S2P6 are 493.9, 510.2, 157.0, and 207.4 ng/mL, respectively (Figure 2c). As for the trispecific antibody, it shows extremely low activity against the antigen, with an EC_50_ of 56.95 μg/mL for hACE2, 17.16 μg/mL for the S protein, and 74.69 μg/mL for XBB RBD (Figure 2a–c). In summary, the original monoclonal antibodies have higher activity than recombinant bsAbs. Among these recombinant antibodies, the bsAbs possess a strong binding activity to their corresponding antigens, where the EC_50_ remains at the ng/mL level. However, the recombinant trispecific antibody only has weak binding activity, with the EC_50_ remaining at the μg/mL level. The changing position of the variable regions seems to influence the binding activity of antibodies. The variable region, when positioned identically to the native monoclonal antibody, exhibits a superior binding activity compared to when it is linked as a scFv to the end of the heavy chain, as evidenced by comparing the EC_50_ of antibodies whose variable regions are designed at different locations within the same antigen group (Figure 2a–c).

### 3.3. The Recombinant Antibodies Exhibit a Superior Protective Efficacy in Pseudoviruses Neutralization Assay

To determine the neutralization activity and protective efficacy at the cellular level of all recombinant antibodies, we carried out neutralization tests with various SARS-CoV-2 pseudovirus subvariants, such as Wuhan, Beta, Delta, BA.2, BA.5, and XBB. The half-maximum inhibitory concentration (IC_50_) values were calculated, and the one-way ANOVA was conducted to assess whether there was significant difference among the antibodies to the different subvariants. The one-way ANOVA results indicated significant difference with the *p* < 0.0001. The recombinant trispecific antibody exhibited superior neutralization activity and broad protective efficacy compared to other bsAbs and native monoclonal antibodies (*p* < 0.0001), with IC_50_ values ranging from 5 to 27 ng/mL for different subvariants, which are significantly lower than those of other recombinant antibodies (Figure 3a–f). Among bispecific antibodies (bsAbs), F-S309 + S2P6 demonstrates superior and broader efficacy compared to others against various SARS-CoV-2 subvariants (*p* < 0.0001), with IC_50_ values of 8, 14, 43, 32, 85, and 23 ng/mL, respectively (Figure 4a–f). The bsAbs F-S2P6 + S309, F-h11B111 + S309, and F-S309 + h11B1 exhibit a broader protective spectrum against diverse subvariants. The IC_50_ of the bsAb F-S2P6 + S309 is 10, 27, 50, 104, 112, and 32 ng/mL against different subvariants (Figure 4a–f). Meanwhile, the IC_50_ of the bsAb F-h11B11 + S309 is 19, 73, 30, 130, 150, and 73 ng/mL, respectively (Figure 4a–f). And the F-S309 + h11B11 shows a similar trend as the previous bsAbs, with IC_50_ values of 15, 41, 20, 137, 116, and 64 ng/mL (Figure 4a–f). However, they do not present lower IC_50_ values compared to F-S309 + S2P6. The bsAbs F-h11B11 + S2P6 and F-S2P6 + h11B11 do not exhibit a similar tendency to other bsAbs (Figure 5). Although F-S2P6 + h11B11 possesses a good neutralization activity against Wuhan and Beta subvariants with IC_50_ values of 43 and 60 ng/mL (Figure 4a,b), its efficacy is not equivalent against the Delta, BA.1, BA.5, and XBB, which are classified as variants of concern (VOC), with the IC50 values of 333, 358, 327, and 591 ng/mL, respectively. The bispecific antibody F-h11B11 + S2P6 displays a distinctive characteristic in its neutralization profile, with the weakest activity observed against the XBB variant, reflected by an IC_50_ value of 684 ng/mL (Figure 4f). In contrast, the IC_50_ values for its neutralizing activity against the other variants are 283, 283, 374, 409, and 372 ng/mL, respectively (Figure 4a–e). Generally speaking, the bsAb F-h11B11 + S2P6 demonstrates the least ideal neutralization activity among the evaluated bsAbs since its IC_50_ values against all subvariants are relatively higher than those of all other bsAbs. Compared to native monoclonal antibodies S2P6 and h11B11, all these bsAbs display better neutralization activity with lower IC_50_ values (*p* < 0.0001), indicating a superior protective efficiency compared to native monoclonal antibodies (Figure 5). S309 exhibits a different scene. The bsAb F-S2P6 + S309, F-h11B111 + S309, F-S309 + S2P6 and F-S309 + h11B11 show a better neutralization activity than S309 against all subvariants with much lower IC_50_ values. Meanwhile, the bsAb F-S2P6 + h11B11 merely exhibits lower IC_50_ values against the Beta and BA.5 subvariants in comparison to S309. The bsAb F-h11B11 + S2P6 possesses a superior neutralization activity with lower IC_50_ values exclusively against the BA.5 subvariant. Additionally, in contrast to the original monoclonal antibody S309, F-h11B11 + S2P6 shows an inferior neutralization activity against other subvariants with significantly higher IC_50_ values.

Unlike the ELISA assay, the positions of variable regions seem to have no influence on the protective efficiency in vitro. The IC_50_ values of most recombinant antibodies show no difference against all variants. The F-S2P6 + h11B11 and F-h11B11 + S2P6 are against Wuhan and Beta variants, a distinction is shown with lower IC_50_ values, while against BA.2, BA.5, XBB variants with much higher IC_50_ values (Figure 5). The paradoxical results imply more explorations into what plays a vital role in the function of antibodies.

### 3.4. The Recombinant Antibodies Confer Protection to Mice Against XBB Infection

To assess the protection at the individual level, we conducted a virus challenge test on BALB/c mice. All mice were intranasally infected with 5 × 10^5^ median tissue culture infectious dose (TCID_50_) of the XBB.1.16 subvariant. 12 h after the infection, recombinant bsAbs, trispecific antibody, the native antibodies, and the control IgG (an antibody targeted at hemagglutinin) were injected into mice at a dose of 10 mg/kg intraperitoneally. 72 h after the infection, lungs were collected and the virus loads were tested by qPCR. Compared with the control IgG, all antibodies in this study showed a therapeutic effect, resulting in a significant reduction in viral loads.

The one-way ANOVA analysis revealed statistically significant differences among the experimental groups (*p* < 0.0001). To further explore these differences, Dunnett’s multiple comparisons test was conducted to determine significant differences between each treatment group and the control group (Figure 6a), while Tukey’s multiple comparisons test was applied to assess significant differences among the treatment groups themselves (Figure 6b). Among the recombined antibodies, the trispecific antibody F-h11B11 + S2P6 + S309 and bsAbs F-S2P6 + S309, F-S309 + S2P6 exhibit the most outstanding protective efficacy due to the lowest viral loads detected in the lungs (Figure 6a). Compared to control IgG group, the viral load detected in the lung of F-h11B11 + S2P6 + S309 group reduced 3.53-fold (*p* < 0.0001), the F-S2P6 + S309 group displayed a reduction in 3.16-fold viral load (*p* < 0.0001), the F-S309 + S2P6 exhibited a decrease in 3.34-fold viral load (*p* < 0.0001). Furthermore, statistical analysis indicated no significant difference in virus titers between the trispecific antibody and bsAbs F-S2P6 + S309, F-S309 + S2P6 (Figure 6b), which indicated changing positions of variable regions would not influence the protective efficiency between F-S2P6 + S309 and F-S309 + S2P6. The other bsAbs F-h11B11 + S2P6, F-h11B11 + S309, F-S2P6 + h11B11, and F-S309 + h11B11 manifested an inferior protective efficiency for the higher viral loads detected in the lungs (Figure 6a). The F-S2P6 + h11B11 displayed no significant difference from F-h11B11 + S2P6 or F-S309 (Figure 6b), again, supporting the previous conclusion that the placement of variable regions would not reduce the effect of bispecific antibodies. Due to the relatively mild virulence of the XBB.1.16 strain in mice, no significant weight loss was observed.

Overall, the bsAbs exhibited an efficient protective efficacy against SARS-CoV-2. Nevertheless, diverse designs of bsAbs yield distinct outcomes. The most effective bsAb designs in this study are F-S2P6 + S309 and F-S309 + S2P6, as well as the trispecific antibody F-h11B11 + S2P6 + S309.

### 3.5. The Recombinant Antibodies Display a Good Affinity to the Target Proteins

To identify the binding affinity of all recombinant antibodies to the target antigens, we measured the KD values by surface plasmon resonance (SPR). The KD values of recombinant antibodies ranged from 9.87 nM to 1460 nM (Figure 7). The recombinant antibodies all exhibited good binding affinity except the F-S2P6 + S309, whose KD values to the antigens reached 300 nM (Figure 7l) and 1460 nM (Figure 7h). Especially, the trispecific antibody F-h11B11 + S2P6 + S309 had good affinity to all three antigens (Figure 7e,j,o).

The results of SPR further enhanced the conclusion that the antibodies designed in this study could provide protective efficiency in vivo and in vitro.

## 4. Discussion

The outbreak of COVID-19 since the end of 2019 has caused tremendous loss of life and property. However, apart from SARS-CoV-2, coronaviruses such as SARS-CoV and MERS-CoV have also brought about global pandemic crises. Now we are confronted with the rapid emergence of many subvariants, including BA.1, BA.2, EG.5, and JN.1 [44]. There is no guarantee that another pandemic caused by coronaviruses will not occur in the future, and, currently, we do not have an ideal medicine for it. Monoclonal antibodies are crucial for anti-virus treatments since they possess specificity and efficiency in protecting individuals from infection. Nevertheless, the continuous gene mutations occurring in the viral genome have rendered existing monoclonal antibodies ineffective. The demand for broad-spectrum antibodies remains urgent.

The multispecific antibodies possess the ability to bind multiple distinct antigens within a single molecule, presenting great potential for development as an anti-virus medicine. Compared to cocktail therapy, they have the advantages of low cost and small dose requirements. The multispecific antibodies targeting SARS-CoV-2 have been studied previously, demonstrating good protective efficiency and neutralization activities [31,45,46]. Here, we designed several bispecific antibodies and one trispecific antibody incorporating Fabs segments of three distinct mechanisms, to achieve a broad-spectrum protective efficacy. The in vivo and in vitro tests confirmed that the trispecific antibody F-h11B11 + S3P6 + S309 and the bsAb F-S2P6 + S309 have the highest efficiency among these antibodies in the study, suggesting the potential of broad-spectrum antiviral efficacy. In this study, we assessed whether modifying the variable regions would impact the functions of bsAbs. It appears that the position would not significantly affect the binding activity and protective efficacy in vivo or in vitro, as the EC_50_ values and IC_50_ values of those pairs of bsAbs, whose variable regions were swapped, are comparable.

However, there are several limitations in our study. One of them is that we did not complete more subvariants of SARS-CoV-2 in the virus challenge test, which might undermine the confidence of recombined antibodies in broad-spectrum anti-virus efficiency, and it is essential to monitor a comprehensive set of indicators to rigorously validate the efficacy of antibodies. In this study, we deployed pseudoviruses of SARS-CoV-2 subvariants for neutralization assays. While pseudovirus systems offer advantages in terms of biosafety and standardization, they present certain limitations compared to authentic virus-based assays. The pseudoviruses platform primarily evaluates the viral entry process mediated by the spike protein, potentially overlooking other aspects of viral replication and pathogenesis observed in live virus systems. Although previous studies have indicated that BALB/c mice are sensitive to Omicron infection, potentially due to mutations in SARS-CoV-2, the absence of hACE2 expression in BALB/c mice raises the uncertainty of the efficacy of h11B11 in this model. Without hACE2, the mechanism of action of h11B11, which specifically targets hACE2, remains unclear in BALB/c mice.

The changing positions of variable regions seems to have had a minor influence on the biological activity of bsAbs in this study, according to the neutralization tests and in vivo experiment. Comparing different positions of variable regions of bsAbs, we found that the determinant of the biological activity of antibodies is the variable regions themselves rather than the positions. The results of ELISA and neutralization tests are not completely consistent. The native antibodies have the highest binding activities to their antigens, while the bsAbs and the trispecific antibodies have the lowest IC_50_ values. We hypothesize that this discrepancy may arise from the fundamental differences between the two assays. The ELISA assay solely measures the antigen-binding capacity of antibodies, whereas neutralization tests evaluate the functional ability to inhibit viral replication. Given their engineered structures, bispecific and trispecific antibodies exhibit more complex structures compared to natural antibodies. This structural complexity may affect their antigen-binding efficiency, potentially leading to higher EC_50_ values in ELISA assays. However, these engineered antibodies possess multiple antiviral mechanisms of action, which could explain their enhanced neutralization potency and, consequently, lower IC_50_ values in neutralization assays. In addition, the specific antibodies and trispecific antibody displayed lower IC_50_ values or lower viral load in lung in this study. Thus, the precise mechanism of these recombinant antibodies needs to be explored further. Do the distinct variable regions with different antiviral mechanisms in the same bispecific antibody play cooperatively through synergism, or do they operate independently through respective mechanistic pathways? The mechanism of this situation requires further exploration.

In summary, this study demonstrates that the designs of combining different mechanisms in bsAb or trispecific antibody are feasible as a therapeutic approach. The bsAb and trispecific antibody indeed enhance the anti-virus efficiency compared to the original antibodies. Among the recombinant antibodies developed in this study, the trispecific antibody demonstrated superior antiviral efficacy in both in vivo and in vitro experiments. This finding suggests that the multi-mechanism design approach holds significant promise for antiviral applications, potentially representing a more effective therapeutic strategy. Regarding bispecific antibody designs, our results revealed that antibodies targeting distinct epitopes on the S protein exhibited enhanced antiviral activity compared to those targeting the host receptor. This observation has redirected our research focus toward viral protein targeting as a preferred strategy for future antibody development. Since we developed therapeutic antibodies in the study, there are some potential modifications that could be added to the antibodies to improve their efficiency, such as modifications to the Fc region to extend half-life or enhance antibody dependent cell-mediated cytotoxicity (ADCC) and antibody dependent cell-mediated phagocytosis (ADCP). These potencies require further research in the future.

Ultimately, we selected three recombined antibodies (F-S2P6 + S309, F-S309 + S2P6, F-h11B11 + S3P6 + S309) with the best broad-spectrum anti-virus efficiency, which are capable of being supplements to the treatment measure of COVID-19.

## 5. Conclusions

Our study has successfully developed recombinant multispecific antibodies with the design containing different antiviral mechanisms. The protective efficacy of these recombinant multispecific antibodies was rigorously evaluated through both in vitro and in vivo experiments, confirming their therapeutic potential. It’s a new supplement to the treatment of COVID-19 and would be an option for clinical. Furthermore, our study explored the impact of positional changing on bispecific antibody efficacy, revealing that spatial configuration does not significantly influence the performance of bispecific antibodies. Although our study has confirmed the efficiency of recombinant multiple-specific antibodies, the precise antiviral mechanism still needs further research. Specifically, it remains to be determined whether the variable regions with different antiviral mechanisms in the same bispecific antibody molecule function synergistically or maintain independent antiviral capacities.

## Figures and Tables

**Figure 1 vaccines-13-00255-f001:**
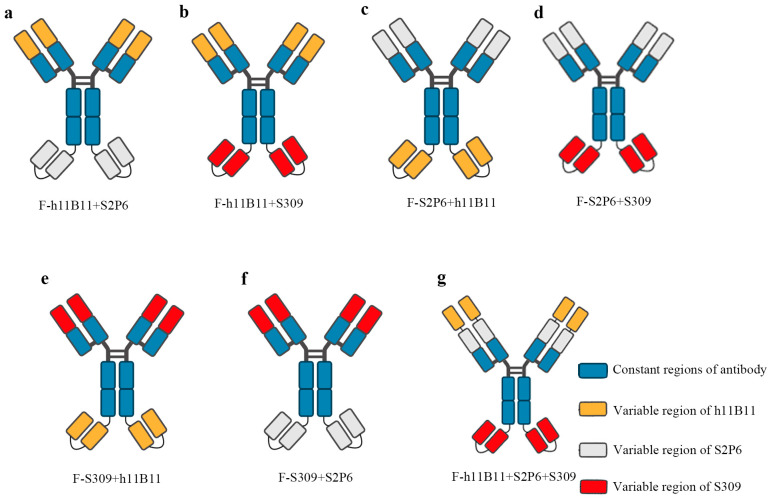
The structure of recombinant antibodies, including bispecific antibodies and trispecific antibody. (**a**–**f**) The bsAb designed based the IgG-scFv format and display the changing position of different Fab fragments design. (**g**) The design of trispecific containing three Fab fragments.

**Figure 2 vaccines-13-00255-f002:**
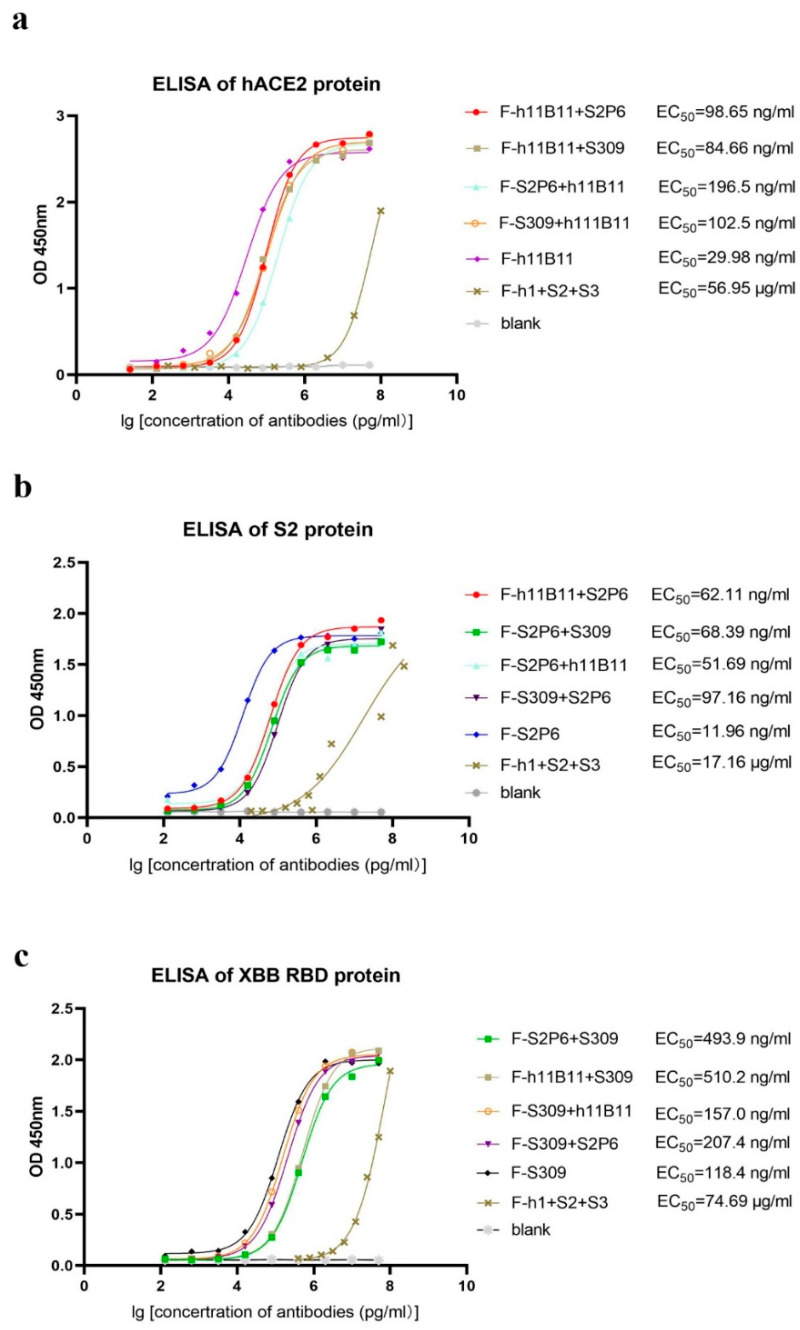
The binding activity of each antibody. The F-h1 + S2 + S3 in this figure stands for F-h11B11 + S2P6 + S309. (**a**) ELISA of antibodies targeted hACE2. Antibodies containing Fab fragments of h11B11 were tested. (**b**) ELISA of antibodies targeted S protein of SARS-CoV-2-Wuhan. Antibodies containing Fab fragments of S2P6 were tested. (**c**) ELISA of antibodies targeted XBB RBD. Antibodies containing Fab fragments of S309 were tested.

**Figure 3 vaccines-13-00255-f003:**
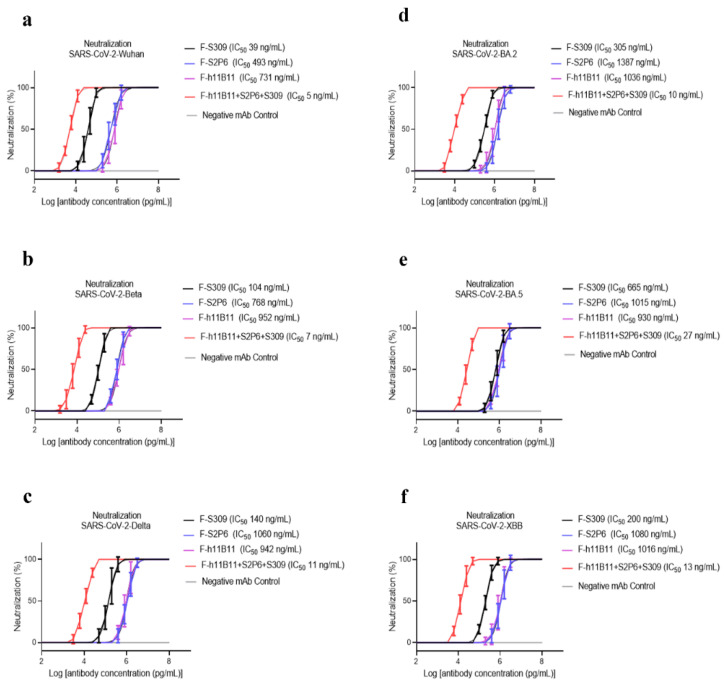
The protective efficiency of trispecific antibody in vitro. (**a**–**f**) The pseudoviruses neutralization test of trispecific antibody against different variants of SARS-CoV-2, which are Wuhan, Beta, Delta, BA.2, BA.5, XBB, respectively. The error bars in the figure means standard deviation. The experiments were repeated 5 times in each dilution fold (*n* = 5).

**Figure 4 vaccines-13-00255-f004:**
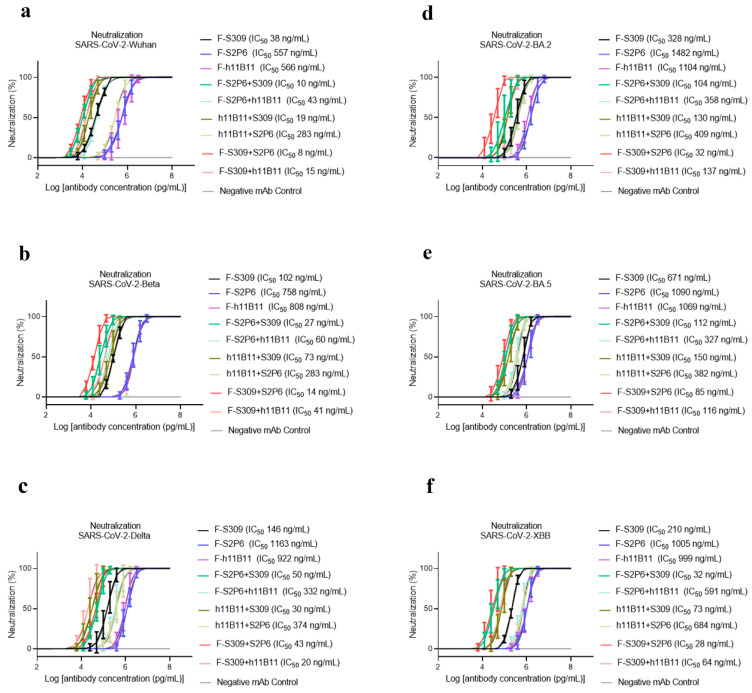
The protective efficiency of bsAbs in vitro. (**a**–**f**) The pseudoviruses neutralization test of bsAbs against different variants of SARS-CoV-2, which are Wuhan, Beta, Delta, BA.2, BA.5, XBB, respectively. The error bars in the figure mean standard deviation. The experiments were repeated 5 times in each dilution fold (*n* = 5).

**Figure 5 vaccines-13-00255-f005:**
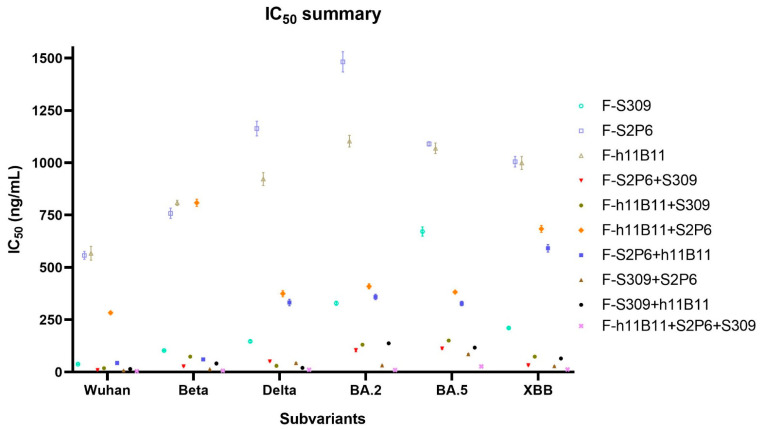
The summary of neutralization test. The summary of IC_50_ values of each antibody against different subvariants of SARS-CoV-2. The error bars in the figure mean standard deviation.

**Figure 6 vaccines-13-00255-f006:**
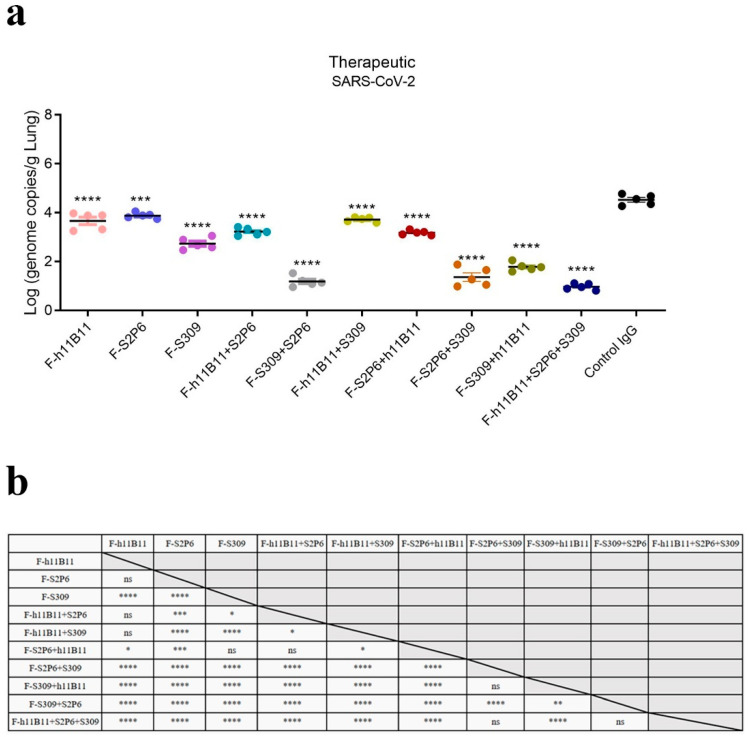
The protective efficiency in vivo. (**a**) The virus loads detected in lung of each mice group (**** *p* < 0.0001, *** *p* = 0.0002 vs. Control IgG). (**b**) The results of Tukey’s multiple comparisons test between each antibody treatment group (**** *p <* 0.0001, *** *p <* 0.001, ** *p <* 0.01, * *p <* 0.05, ns = not significant).

**Figure 7 vaccines-13-00255-f007:**
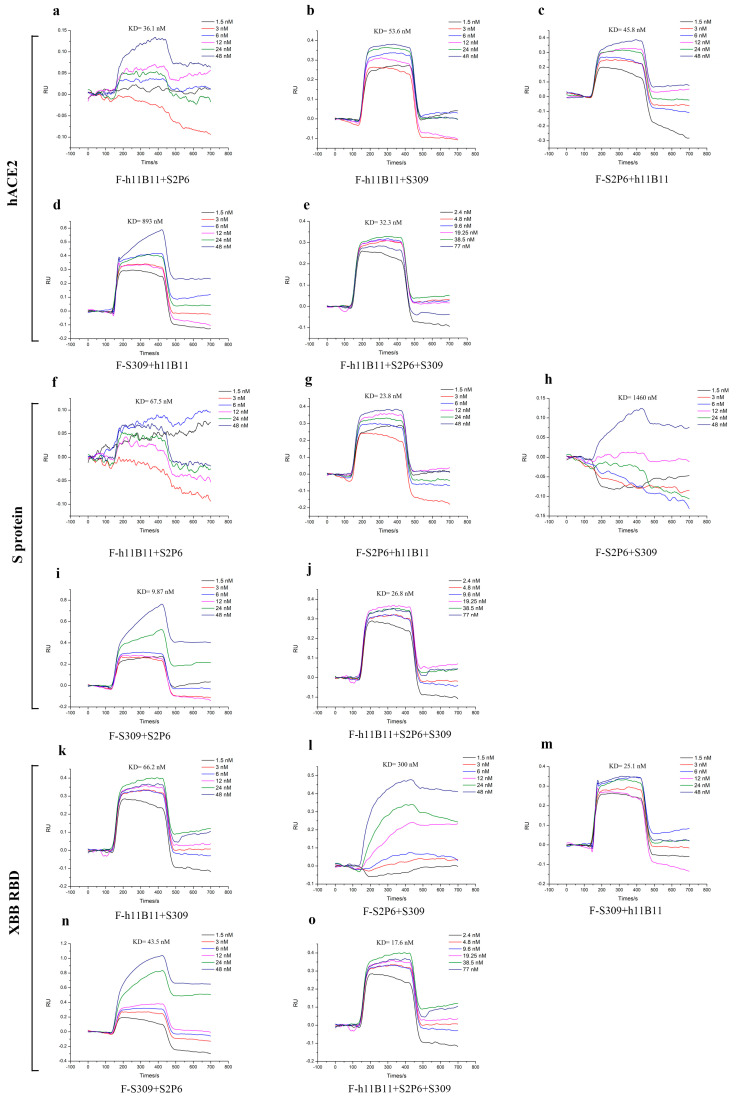
The binding affinity of recombinant antibodies through SPR analysis. (**a**–**e**) The antibodies targeted hACE2 with different concentrations. (**f**–**j**) The antibodies targeted S protein of SARS-CoV-2-Wuhan with different concentrations. (**k**–**o**) The antibodies targeted XBB RBD with different concentrations.

## Data Availability

The data presented in this study are available on request from the corresponding author.

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
