# Peer review of "Design and Characterization of Bispecific and Trispecific Antibodies Targeting SARS-CoV-2"

_vaccines, 2025, doi:10.3390/vaccines13030255_

Round 1
Reviewer 1 Report
Comments and Suggestions for Authors
I have read carefully and with curiosity the article titled "Design and Characterization of Bispecific and Trispecific Antibodies Targeting SARS-CoV-2" by Wang et al. The study is well designed. Both the introduction and the discussion are well prepared and the conclusions are clear and concise.
The methods are well described and the results are
have shown very well.
I am not a native speaker of English, so I don't feel I can decide whether it is good or not in English, but it sounds to me like it is well written and developed.
Author Response
We sincerely appreciate your time and effort in reviewing our manuscript. We are grateful for the constructive comments and suggestions, which have significantly improved the quality of our paper.
Reviewer 2 Report
Comments and Suggestions for Authors
Topical paper showing advantage of bi and trivalency in targeting coronaviruses, but unsurprising and unremarkable conclusions given nature’s preference for polyclonal antibodies in tackling viruses
Specifics:
Figure 1, Clear diagrams make the construct designs easy to understand, thank you
Figure 2 legend, Check English in ‘Those antibodies contain…’ sentences. Also make it clearer what F-h.1+S2+S3 is.
Figures 3 and 4, Strange and confusing to use reverse scales for the x axes
Figure 5, Not sure this should be a line graph, suggest unlinked points
308, Honest assessment of limitations
311, Unsurprising that the native antibodies exhibit the highest binding
313, Not sure mechanism requires further exploration, as it will almost certainly be occlusion and compromised access
Author Response
Reviewer #1, Comment 1:
Figure 1, Clear diagrams make the construct designs easy to understand, thank you.
Response:
Thanks for the suggestions. We have added legends with different colors to indicate the variable regions of different antibodies in the Figure 1 (Page 12).
Reviewer #1, Comment 2:
Figure 2 legend, Check English in ‘Those antibodies contain…’ sentences. Also make it clearer what F-h.1+S2+S3 is.
Response:
We are sorry for the negligence about the language. We have revised the language in the legend of Figure 2 and added a clear explanation about what “F-h1+S2+S3” stands for (Page 15).
Reviewer #1, Comment 3:
Figures 3 and 4, Strange and confusing to use reverse scales for the x axes.
Response:
We agree with the reviewer's suggestion. We have adjusted the x axes options to make it a standard orientation for better understanding (Page 18-19).
Reviewer #1, Comment 4:
Figure 5, Not sure this should be a line graph, suggest unlinked points.
Response:
We appreciate the reviewer's suggestion about the Figure 5. We have carefully considered the comment and agree that the choice of graph type is important for clarity. However, in this case, we believe that retaining the lines connecting the data points is necessary to illustrate the trends in IC50 values against different virus subvariants for each antibody. Removing the lines might make it difficult to visualize the overall patterns and comparisons, especially given the large number of data points.
Reviewer #1, Comment 5:
308, Honest assessment of limitations. 311, Unsurprising that the native antibodies exhibit the highest binding.
Response:
Thanks for the suggestions about the discussion. We apologize for the lack of depth in the discussion section. We have added more discussion content about the results displayed in ELISA assay and neutralization assay, including probe into why the native antibodies showed a highest binding activity and why the recombinant antibodies have lower IC50 values (Line 436-446, Page 27). Meanwhile, we have added more limitations in our study in the discussion section (Line 420-427, Page 26).
Reviewer 3 Report
Comments and Suggestions for Authors
Wang JY and colleagues used constructed several bispecific and trispecific antibodies based on currently available monoclonal antibodies (S2P6, S309 and h11B11). Their aim was to simultaneously targeting human ACE2 protein, SARS-CoV-2 viral spike and spike-RBD epitopes to achieve potent and broaden viral neutralizing activity. They characterized binding activities of antibodies and target antigens, viral neutralization activity was evaluated using pseudoviruses; further the authors tested and compared the treatment efficacy of these antibodies for the recent SARS-CoV-2 variant XBB 1.16 in mice. Due to virus mutation, antiviral monoclonal antibody, such as S2P6 and S309 gradually loss their potency and therapeutic value. The topic of this study is of importance and significance in developing effective prophylactic and therapeutic antibodies against all viral infectious diseases. However, there are some important issues have to be addressed.
Major comments:
- The design and construction of those bispecific and tri-specific antibodies seem out of the blue, the authors only provided references for the isolation of those original monoclonal antibodies without giving details of their own work, for instance, the source of variable genes for each antibody, the plasmid that expressing constant region of heavy and light chain.
- Neutralization assay these the bispecific and trispecific antibodies was performed only using SARS-CoV-2 pseudoviruses. This should also be done with live viruses.
- In vivo protection potency of these antibodies was only performed against one virus XBB 1.16, other recent viral variants should be tested. The animal experiment was only observed for 3 days after virus infection, which is not sufficient. Longer time points should be studied. In addition, evaluation of protective effects against viral infection, merely determination of lung viral load is not sufficient, control of disease severity (such as body weight changes) and protection against lung tissue histopathological damage should be evaluated.
- Since bispecific antibody and trispecific antibody were explored in this study, it is not clearly indicated or suggested in discussion which strategy is superior and why; what further potential modification can also be done to improve the function of bi-, trispecific antibody? These should be discussed.
- As there are already reports using similar strategy to construct bispecific antibody using the same monoclonal antibody, h11B11, S309 and S2P6 (https://doi.org/10.1080/22221751.2024.2404166), what is the advancement the authors made comparing to previous report.
- Explain why using BALB/c mice for in vivo protection evaluation, but not K18-hACE2 mice or Syrian hamster models.
Minor comments:
- Statistics section was not included in the manuscript.
- Materials and Methods section: under subtitle “Therapeutic efficacy test in mice” there is no descriptions for animals used in the study.
- In line 167-168, 5x10^5TCID50 XBB.1.16 in 75mL DMEM should be wrong, please check and correct; giving virus to mice usually use the term “inoculate” instead of “transduce”.
- For mice study, why is the dose of 10mg/kg of antibody was chosen? What is the route of injection? What is reason to give the antibodies 12 hours after virus infection? Were these antibodies still effective if given at 24 hour and 48 hours after virus infection?
- Figure 1 should be labeled clearly which V region is from which antibody and indicated in its legend.
- In Figure 2, what “F-h1-S2-S3” represents? In figure 2B, S protein of which virus strain was used to coat ELISA plate? this should be stated in the figure legend. Figure 2 A, B, C, the curve labels are very hard to read, may consider to make them consistent for all charts, i.e. same antibody with same label
- The authors tested binding activity of their recombinant antibodies to hACE2, S protein and XBB S-RBD with ELISA. Please explain how this assay reflects antibody binding affinity and avidity?
- The possible reason that lead to the observed lower binding activity in ELISA whereas better neutralization activity for bispesific and trispecific antibodies should be discussed.
- Neutralization assay using pseudoviruses should be stated in Abstract, results and the legends for figure 3 and 4.
- In figure 6, which SARS-CoV-2 gene fragment was determined for lung viral load? What was the statistical analysis method was used? The stars labeled in figure 6A are confusing.
There are some errors in the manuscript should be carefully checked and correct.
Author Response
Reviewer #2, Comment 1:
The design and construction of those bispecific and tri-specific antibodies seem out of the blue, the authors only provided references for the isolation of those original monoclonal antibodies without giving details of their own work, for instance, the source of variable genes for each antibody, the plasmid that expressing constant region of heavy and light chain.
Response:
We sincerely appreciate the reviewer's valuable suggestion. We have added detailed information regarding the method used to acquire the variable region sequences of each antibody, along with the peptide linker sequence and the expression vector used in this study (Lines 154-158, Pages 7-8). Additionally, we have cited relevant reference (Reference #42, Line 155, Page7) about relevant methods. We hope these revisions address the reviewer's concern and hope they meet the reviewer's expectations.
Reviewer #2, Comment 2:
Neutralization assay these the bispecific and trispecific antibodies was performed only using SARS-CoV-2 pseudoviruses. This should also be done with live viruses. In vivo protection potency of these antibodies was only performed against one virus XBB 1.16, other recent viral variants should be tested.
Response:
We sincerely appreciate your valuable advice. We fully concur with your perspective. It is indeed crucial to test a broader range of authentic SARS-CoV-2 strains and conduct animal experiments using these strains to further validate the efficacy of antibodies. However, our institution cannot perform experiments involving live SARS-CoV-2 virus independently. Therefore, we must collaborate with a partner institution for such experiments, which presents certain limitations. Our current cooperation agreement with the partner institution has expired, and we need to finalize the next agreement before proceeding with these experiments. Consequently, we are unable to conduct further experiments involving live SARS-CoV-2 virus in the short term. We will supplement these experiments when conditions permit. Thank you once again for your suggestion, and we hope for your understanding.
Accordingly, we have added a discussion of this limitation in the revised manuscript (Lines 420-427, Page 26).
Reviewer #2, Comment 3:
The animal experiment was only observed for 3 days after virus infection, which is not sufficient. Longer time points should be studied. In addition, evaluation of protective effects against viral infection, merely determination of lung viral load is not sufficient, control of disease severity (such as body weight changes) and protection against lung tissue histopathological damage should be evaluated.
Response:
We appreciate your valuable suggestions and concur with your views. We also share your enthusiasm for expediting these experiments. Regarding the animal weight data, due to the relatively mild pathogenicity of the XBB.1.16 strain in mice, no significant weight loss was observed. This information has been included in the results section. Due to the low virulence of the virus, prolonged observation revealed that the virus was completely cleared from the lungs of all infected mice.
As previously mentioned, our current resources do not permit the immediate initiation of new SARS-CoV-2 animal infection studies. We regret this limitation and will endeavor to conduct these experiments at the earliest opportunity when conditions allow. We have also added the related information in the discussion section, “and it is essential to monitor a comprehensive set of indicators to rigorously validate the efficacy of antibodies”. We kindly request your understanding in this matter.
Reviewer #2, Comment 4:
Since bispecific antibody and trispecific antibody were explored in this study, it is not clearly indicated or suggested in discussion which strategy is superior and why; what further potential modification can also be done to improve the function of bi-, trispecific antibody? These should be discussed.
Response:
Thanks for reviewer’s suggestion so that we could notice the inadequacy in the paper. We have added more detail in the discussion section about what strategy could be better concluded by this study and the potential modifications to improve the antibodies efficiency (Line 451-466, Page 27-28). We hope this revised edition may fulfill your concern.
Reviewer #2, Comment 5:
As there are already reports using similar strategy to construct bispecific antibody using the same monoclonal antibody, h11B11, S309 and S2P6 (https://doi.org/10.1080/22221751.2024.2404166), what is the advancement the authors made comparing to previous report.
Response:
We sincerely appreciate the reviewer's comment and the reference to previous studies using similar strategies to construct bispecific antibodies. Compared to previous study, our study has some advantages: we explored the design of tri-specific antibody and verify its efficiency in vivo and in vitro, which provide a new way to multiple specific antibodies. The bispecific antibodies we designed contained different variable regions with different mechanisms that included target RBD, S2 subunit and host receptor rather than only target the viral protein. The divers combinations explored more possibilities and supplement to the treatment to COVID-19. In summary, our study not only builds upon the previous study, but also provided novel insights to the design and application of multiple specific antibodies. We believe our work contribute to the field and clinical development. Thank you again for the valuable feedback.
Reviewer #2, Comment 6:
Explain why using BALB/c mice for in vivo protection evaluation, but not K18-hACE2 mice or Syrian hamster models.
Response:
Thanks for the question about the animal model. Based on previous studies[1–3], the BALB/c mice were sensitive to Omicron infection, which illustrated BALB/c could be the animal model for Omicron infection experiment. Meanwhile, we have added a new reference in the Materials and methods sections under the subtitle “Therapeutic efficacy test in mice” to explain why we chose BALC/c as animal model (Reference #43, Line 211, Page 10).
Reviewer #2, Comment 7:
Statistics section was not included in the manuscript.
Response:
Thanks for the reviewer’s advice. We have added statistic analysis part in the Materials and Methods section to state what statistical method and software was used in the study (Line 226-229, Page 11). Meanwhile, we have reiterated the statistical method in the result parts of in vivo test (Line 353-359, Page 21).
Reviewer #2, Comment 8:
Materials and Methods section: under subtitle “Therapeutic efficacy test in mice” there is no descriptions for animals used in the study.
Response:
We appreciate reviewer pointed out the inadequacy of our paper in the Materials and Methods section. We have added clear descriptions to the experimental animals at the under subtitle “Therapeutic efficacy test in mice” and added one relevant reference to illustrate why we chosen BALB/c for the in vivo test (Line 209-211, Page 10).
Reviewer #2, Comment 9:
In line 167-168, 5x10^5 TCID50 XBB.1.16 in 75mL DMEM should be wrong, please check and correct; giving virus to mice usually use the term “inoculate” instead of “transduce”.
Response:
Thanks for the suggestions about our language errors. We have fixed the wrong phrases and adopted your advice replacing the word “transduce” to “inoculate” (Line 217-219, Page 10). Thanks again for your patience.
Reviewer #2, Comment 10:
For mice study, why is the dose of 10mg/kg of antibody was chosen? What is the route of injection? What is reason to give the antibodies 12 hours after virus infection? Were these antibodies still effective if given at 24 hour and 48 hours after virus infection?
Response:
Due to experimental limitations, we conducted an antibody protection study with only a single dosage. Based on our previous research on viral neutralizing antibodies (https://pmc.ncbi.nlm.nih.gov/articles/PMC10286687/), a dose of 10 mg/kg has been shown to provide effective protection in animal models; therefore, we selected this dosage. The antibodies and control IgG (an antibody targeting hemagglutinin) were administered via intraperitoneal injection, this information has been included in the methodology section. Given the low virulence of the virus, extended observation demonstrated complete clearance of the virus from the lungs of all infected mice. Consequently, antibody treatment was administered 12 hours post-viral infection.
Reviewer #2, Comment 11:
Figure 1 should be labeled clearly which V region is from which antibody and indicated in its legend.
Response:
Thanks for the suggestions. We have added legends with different colors to indicate the variable regions of different antibodies in the Figure 1 (Page 12).
Reviewer #2, Comment 12:
In Figure 2, what “F-h1-S2-S3” represents? In figure 2B, S protein of which virus strain was used to coat ELISA plate? this should be stated in the figure legend.
Response:
We sincerely appreciate the reviewer's comment on the Figure 2. We have added a clear explanation about what “F-h1+S2+S3” stands for in the figure legend and which virus strain was used to coat ELISA plate (Page 15).
Reviewer #2, Comment 13:
Figure 2 A, B, C, the curve labels are very hard to read, may consider to make them consistent for all charts, i.e., same antibody with same label.
Response:
We appreciate the reviewer's comment about the labels in Figure 2. We agree that the current curve labels are not optimal and have revised the figure to ensure clarity. We have changed the labels to make sure the same label for each antibody across all panels (Fig. 2 a-c, Page 15). Meanwhile, we have reduced the sized of each data point to make the curve more visible.
Reviewer #2, Comment 14:
The authors tested binding activity of their recombinant antibodies to hACE2, S protein and XBB S-RBD with ELISA. Please explain how this assay reflects antibody binding affinity and avidity?
Response:
We appreciate the reviewer's question about the interpretation of ELISA data in terms of antibody binding affinity and avidity. In our study, the ELISA assay was used to evaluate the binding activity of our recombinant antibodies. The binding affinity of the antibodies was indirectly assessed by determining the half-maximal effective concentration (EC50) from the dose-response curves. Lower EC50 values indicate higher binding affinity. In our study, compared to recombinant antibodies the native antibodies had the lowest EC50 values besides the EC50 values of recombinant antibodies remained in ng/mL to μg/mL (Fig 2, Page 15). To further evaluate the binding affinity, we have supplemented our study with Surface Plasmon Resonance (SPR) experiments, which provide direct measurements of binding affinity (Fig 7) (Line 374-385, Page 22-24).
Reviewer #2, Comment 15:
The possible reason that leads to the observed lower binding activity in ELISA whereas better neutralization activity for bispesific and trispecific antibodies should be discussed.
Response:
Thanks for the suggestions about the discussion. We apologize for the lack of depth in the discussion section. We have added more discussion content about the reason leads to inconsistent results of ELISA assay and neutralization assay. (Line 436-446, Page 27).
Reviewer #2, Comment 16:
Neutralization assay using pseudoviruses should be stated in Abstract, results and the legends for figure 3 and 4.
Response:
Thanks for reviewer’s suggestion focus on details. We have added pseudoviruses statement in Abstract (Line 33, Page 2), results (Line 286, Page 16) and the legends of Figure 3 and 4 (Page 18-19). Thanks again for your patience.
Reviewer #2, Comment 17:
In figure 6, which SARS-CoV-2 gene fragment was determined for lung viral load? What was the statistical analysis method was used? The stars labeled in figure 6A are confusing.
Response: N gene fragment from SARS-CoV-2 was determined for lung viral load (We have added the information in Method section).
The one-way ANOVA analysis revealed statistically significant differences among the experimental groups (p < 0.0001). To further explore these differences, Dunnett's multiple comparisons test was conducted to determine significant differences between each treatment group and the control group (Fig. 6a), while Tukey's multiple comparisons test was applied to assess significant differences among the treatment groups themselves (Fig. 6b). We have added this information on the result section.
We have added statistic analysis part in the Materials and Methods section to state what statistical method and software was used in the study (Line 226-229, Page 11). Meanwhile, we have reiterated the statistical method in the result parts of in vivo test (Line 353-359, Page 21). The stars labeled in Figure 6A represented different p values of Dunnett's multiple comparisons test, which determine significant differences between each treatment group and the control group. The legend also states what is the meaning of stars labels (Fig 6, Page 22).
Reviewer 4 Report
Comments and Suggestions for Authors
The authors report the generation and evaluation of different bi-specific and one tri-specific monoclonal antibody to be used as potential therapy in patients with SARS-CoV-2 or related coronaviral infections. The antibodies had lower binding affinity than the corresponding monoclonal antibodies, but had superior neutralizing activity in vitro and in an in vivo mouse model.
The in vitro data are interesting, although the different outcomes between the ELISA and neutralization test are surprising and unexplained. The neutralization data presented in Figures 3 – 5 would benefit from statistical analysis to back-up statements such as “Only when the F-S2P6 + h11B11 and F-h11B11 + S2P6 are against Wuhan and Beta variants, a distinction is shown, while there is no difference against BA.2, BA.5, and XBB variants” (lines 257-259). The mouse experiments were performed with BALB/c mice (line 263) which, unfortunately, compromises the validity of these experiments as SARS-CoV-2 does not bind to mouse ACE-2 and mice do not support infection and replication of SARS-CoV-2. These experiments should be performed with hACE2-transgenic mice. Without solid in vivo data the manuscript has limited significance.
Other comments:
· Line 76-77. “…..numerous applications of bsAbs in the treatment of cancers and infectious diseases, such as blinatumomab for leukemia, amivantamab for lung cancer.” I am not familiar with commercial bsAbs for infectious diseases, please add a reference to support this statement.
· Line 119. Methods. Please include a paragraph of statistical analysis.
· Line 120 – 134. Please provide a more detailed description of the generation of the recombinant antibodies including spacer sequences, plasmids, etc. The methods should be described in such a way that the work can be replicated. The description should be written in past tense.
· Line 164. Please provide details about the mice (age, sex, strain, number of mice/group). However, these experiments should be performed in hACE2 transgenic mice, not BALB/c mice as mentioned above.
· Figure 1. Please specify which Fab domains correspond with the different colors.
· Figures 3 and 4 are summarized in Figure 5 showing essentially the same data. I would suggest keeping Figure 5 and providing Figures 3 and 4 as supplemental data. The authors should also state how often these experiments were repeated.
· Figure 6. It appears that the statistical analysis was not performed correctly (repeated pair-wise comparisons with Student’s t-test). The authors should evaluate the data using a one-way ANOVA, and, if significant, a post-hoc test with correction for multiple comparisons.
· Line 311. “The results of ELISA and neutralization tests are not completely consistent.” This is an understatement, as the results are quite different. The authors should expand their discussion and provide potential explanations for this difference. The Discussion should also address the effect of different positions of the variable regions on the biological activity (or lack thereof) as mentioned in lines 255-259.
· The references are inconsistent in format and some (e.g., #15 and #43) are incomplete.
Comments on the Quality of English LanguageMostly OK
Author Response
The neutralization data presented in Figures 3 – 5 would benefit from statistical analysis to back-up statements such as “Only when the F-S2P6 + h11B11 and F-h11B11 + S2P6 are against Wuhan and Beta variants, a distinction is shown, while there is no difference against BA.2, BA.5, and XBB variants”
Response:
We appreciate the reviewer's advice on the statements in the neutralization part. We have noticed this unscientific expression without statistical analysis. We have changed the expression in this part to compare the IC50 values rather than a statement about whether there was significant difference. (Line 336-339, Page 20).
Reviewer #3, Comment 2:
The mouse experiments were performed with BALB/c mice (line 263) which, unfortunately, compromises the validity of these experiments as SARS-CoV-2 does not bind to mouse ACE-2 and mice do not support infection and replication of SARS-CoV-2. These experiments should be performed with hACE2-transgenic mice. Without solid in vivo data the manuscript has limited significance.
Response:
Thanks for the question about the animal model. Based on previous studies [1–3], the BALB/c mice were sensitive to Omicron infection, which illustrated BALB/c could be the animal model for Omicron infection experiment. Meanwhile, we have added a new reference in the Materials and methods sections under the subtitle “Therapeutic efficacy test in mice” to explain why we chose BALC/c as animal model (Reference #43, Line 211, Page 10).
Reviewer #3, Comment 3:
“…numerous applications of bsAbs in the treatment of cancers and infectious diseases, such as blinatumomab for leukemia, amivantamab for lung cancer.” I am not familiar with commercial bsAbs for infectious diseases, please add a reference to support this statement.
Response:
We are sorry for the oversight about the statement “numerous applications of bsAbs in the treatment of cancers and infectious diseases”. At present, there are few commercial bsAbs applied to infectious diseases. Most bsAbs targeted infectious disease are remaining research stage not approved listing, just like some bsAbs mentioned later in the abstract. Thus, we have deleted the statement “and infectious diseases” (Line 100, Page 5). We apologize again for the ignorance about the expression.
Reviewer #3, Comment 4:
Methods. Please include a paragraph of statistical analysis.
Response:
Thanks for the reviewer’s advice. We have added statistic analysis part in the Materials and Methods section to state what statistical method and software was used in the study (Line 226-229, Page 11). Meanwhile, we have reiterated the statistical method in the result parts of in vivo test (Line 353-359, Page 21).
Reviewer #3, Comment 5:
Please provide a more detailed description of the generation of the recombinant antibodies including spacer sequences, plasmids, etc. The methods should be described in such a way that the work can be replicated. The description should be written in past tense.
Response:
We sincerely appreciate the reviewer's valuable suggestion. We have added detailed information regarding the method used to acquire the variable region sequences of each antibody, along with the peptide linker sequence and the expression vector used in this study (Lines 154-158, Pages 7-8). Additionally, we have cited relevant reference about methods (Reference #42, Line 155, Page 8). Meanwhile, we have checked the language to ensure written in past tense.
Reviewer #3, Comment 6:
Please provide details about the mice (age, sex, strain, number of mice/group). However, these experiments should be performed in hACE2 transgenic mice, not BALB/c mice as mentioned above.
Response:
We appreciate reviewer pointed out the inadequacy of our paper in the Materials and Methods section. We have added clear descriptions to the experimental animals at the under subtitle “Therapeutic efficacy test in mice” about the age, sex, strain, number of mice in each group (Line 209-212, Page 10). We have added one relevant reference to illustrate why we chosen BALB/c for the in vivo test (Reference #43, Line 211, Page 10). As mentioned above, previous studies [1–3] had used BALB/c in the Omicron infection experiment, which support BALB/c could be the animal model for our study.
Reviewer #3, Comment 7:
Figure 1. Please specify which Fab domains correspond with the different colors.
Response:
Thanks for the suggestions. We have added legends with different colors to indicate the variable regions of different antibodies in the Figure 1 (Page 12).
Reviewer #3, Comment 8:
Figures 3 and 4 are summarized in Figure 5 showing essentially the same data. I would suggest keeping Figure 5 and providing Figures 3 and 4 as supplemental data. The authors should also state how often these experiments were repeated.
Response:
We sincerely appreciate the reviewer's suggestion to streamline the presentation of Figures 3, 4, and 5. However, after careful consideration, we believe that retaining Figures 3 and 4 in the main text is essential as the data in Figure 3 and 4 need to be referenced several times in the text. Keeping Figure 3 and 4 in the text would benefit deader easier to read. At the same time, we have noticed there was no description on how often these experiments were repeated. Thus, we have added the repeat time in the legend of Figure 3 and 4 (Page 18-19).
Reviewer #3, Comment 9:
It appears that the statistical analysis was not performed correctly (repeated pair-wise comparisons with Student’s t-test). The authors should evaluate the data using a one-way ANOVA, and, if significant, a post-hoc test with correction for multiple comparisons.
Response:
We are sorry for the lack of statement of detailed statistical analysis. We have added the process of statistical analysis used in the in vivo test (Line 353-359, Page 21), which includes one-way ANOVA, Dunnett's multiple comparisons test and Tukey's multiple comparisons test. The revision pointed out there are significant differences among the experimental groups by one-way ANOVA analysis (p < 0.0001). Dunnett's multiple comparisons test is used for comparing the difference between each treatment group and the control group. Tukey's multiple comparisons test is sued for comparing difference between each treatment group.
Reviewer #3, Comment 10:
The results of ELISA and neutralization tests are not completely consistent.” This is an understatement, as the results are quite different. The authors should expand their discussion and provide potential explanations for this difference. The Discussion should also address the effect of different positions of the variable regions on the biological activity (or lack thereof) as mentioned in lines 255-259.
Response:
Thanks for the suggestions about the discussion. We apologize for the lack of depth in the discussion section. We have added more discussion content about the results displayed in ELISA assay and neutralization assay, including probe into why the native antibodies showed a highest binding activity and why the recombinant antibodies have lower IC50 values (Line 436-446, Page 27). Also, we have added different positions of the variable regions on the biological activity in the discussion section (Line 428-432, Page 26-27).
Reviewer #3, Comment 11:
The references are inconsistent in format and some (e.g., #15 and #43) are incomplete.
Response:
We sincerely thank the reviewer's careful attention to the formatting and completeness of our references. In response to this comment, we have thoroughly reviewed and revised all references in the manuscript to ensure consistency and completeness by the software Zotero to fulfill the reference format of Vaccine, including the reference #15 and #43 (Line 540-543, Line 645-649).
Round 2
Reviewer 3 Report
Comments and Suggestions for Authors
The author has addressed most of my concerns. However, to the fact that they are not able to perform live-virus experiments due to lack of suitable facilities, I incline to accept this point to be discussed as a limitation of study. There is another important issue was still not be addressed adequately. One of the components in the Tri-specific antibody was h11B11 which targets human ACE2. In vitro neutralizing assay was performed using hACE2-expressing Hela cells, which demonstrated potent neutralizing activity, the activity was also much potent than bispecific antibody containing S309 and S2P6 (in figure 3 and figure 4). While the in vivo data looks to me that Tri-specific antibody F-h11B11+S2P6+S309 had similar protection effect as Bispecific antibody F-S2P6+S309 and S309+S2P6 by viral load reduction in mouse lung (Figure 6a). Considering the host specificity of h11B11, the in vivo study should be performed in K18-hACE2 mice instead of BALB/c mice. Since the authors did not provide data on the interaction of h11B11 with mouse ACE2, the in vivo effect of h11B11 in Tri-specific antibody remains unclear. This is an important point has to be addressed, if not practical by experiments, should be by discussion with relevant references.
Author Response
Thanks for the comments about raising this important point regarding the host specificity of h11B11 and its implications for the in vivo experiment. In the last respond letter, we explained the BALB/c mice were sensitive to the Omicron infection but we did ignore whether the h11B11 would work properly in BALB/c mice. Due to the lack of relevant research on h11B11 in BALB/c mice, we have added some discussion at limitation of our study part (Line 381-385, Page 15).
Reviewer 4 Report
Comments and Suggestions for Authors
The authors addressed most of my previous comments. I have a few remaining issues that should be addressed to improve the quality of the manuscript.
Figures 3 and 4 - please indicate what the error bars indicate (SEM or SD, of how many replicates)?
As I mentioned in my previous comment, Figure 5 provides essentially the same information as Figures 3 and 4. The authors elected to keep all figures. The value of figure 5, which is a summary of the data in Figures 3 and 4, should be enhanced by including error bars of the four independent replicates (according to the updates legends of figures 3 and 4) and statistical analysis. This will strengthen statements in the text that the neutralization activity of certain antibodies is "relatively higher" (line 279), "better" (line 283) or "superior" (line 286).
Comments on the Quality of English LanguageCan be improved
Author Response
Reviewer #4, Comment 1:
Figures 3 and 4 - please indicate what the error bars indicate (SEM or SD, of how many replicates)?
Response:
We sincerely appreciate the reviewer's valuable suggestions regarding the clarity of Figures 3 and 4. We have revised the figure legends to provide more detailed explanations about the error bars (representing standard deviation) and the number of experimental replicates (Page 8, 10).
Reviewer #4, Comment 2:
As I mentioned in my previous comment, Figure 5 provides essentially the same information as Figures 3 and 4. The authors elected to keep all figures. The value of figure 5, which is a summary of the data in Figures 3 and 4, should be enhanced by including error bars of the four independent replicates (according to the updates legends of figures 3 and 4) and statistical analysis. This will strengthen statements in the text that the neutralization activity of certain antibodies is "relatively higher" (line 279), "better" (line 283) or "superior" (line 286).
Response:
Thanks again for the careful inspection and patience. After receiving your comment, we sincerely reconsider the design of Figure 5 and have decided to redraw Figure 5 into a dot figure with standard deviation error bars (Page 11). Meanwhile, we have added the statistical analysis and p values to support the statement in the text that the neutralization experiment (Line 257-261, Page 8, Line 263, Line 266, Page 9, Line 287 Page 10).
Besides, we noticed there were 4 points of clarification for our manuscript in the e-mail. Here, we would like to point out the revision of our manuscript. A Conclusion part has been added to the manuscript (Line 427-439, Page 16). The Abstract part was revised to structured abstract with clear headings (Line 18-34, Page 1). Regarding the main text part, we have added culture conditions of HEK 293F (Line 132-133, Page3), more detailed description about in vivo experiment with p values (Line 319-322, Line 324-325, Line 329-330, Page 11-12), more hypothesis in the Discussion part (Line 401-406, Page 16). The Curriculum Vitae of all authors would be attached to the e-mail, please check. We hope that our revisions have addressed all the concerns raised by the reviewers. Thank you again for your time and effort in improving our manuscript.